# Potential Use of Chitosan-TiO_2_ Nanocomposites for the Electroanalytical Detection of Imidacloprid

**DOI:** 10.3390/polym14091686

**Published:** 2022-04-21

**Authors:** Blanca Estela Castillo, Evgen Prokhorov, Gabriel Luna-Bárcenas, Yuriy Kovalenko

**Affiliations:** Cinvestav del IPN, Unidad Querétaro, Queretaro 76230, Mexico; becr_iq@yahoo.com.mx (B.E.C.); gabriel.luna@cinvestav.mx (G.L.-B.); kovalenko.yuriy@gmail.com (Y.K.)

**Keywords:** chitosan, titanium dioxide, interface layer, dielectric spectroscopy, cyclic voltametric, imidacloprid

## Abstract

The detection of toxic insecticides is a major scientific and technological challenge. In this regard, imidacloprid is a neonicotinoid that is a systemic insecticide that can accumulate in agricultural products and affect human health. This work aims to study the properties of chitosan–TiO_2_ nanocomposites in which nanoparticles with high surface area serve as molecular recognition sites for electroanalytical imidacloprid detection. We show that the best sensitivity to imidacloprid was obtained using a modified electrode with a chitosan–TiO_2_ nanocomposite with a 40 wt.% of TiO_2_ nanoparticles. By using a three-phase effective permittivity model which includes chitosan, TiO_2_, an interface layer between nanoparticles and a matrix, we showed that nanocomposites with 40 wt.% of TiO_2_ the interface volume fraction reaches a maximum. At higher nanoparticle concentration, the sensitivity of the sensor decreases due to the decreasing of the interface volume fraction, agglomeration of nanoparticles and a decrease in their effective surface area. The methodology presented can be helpful in the design and optimization of polymer-based nanocomposites for a variety of applications.

## 1. Introduction

Chitosan (CS)–titanium dioxide nanoparticles (TiO_2_ NPs) composites are one of the most attractive materials which have interesting technological properties and applications. Nanocomposites based upon CS–TiO_2_ NPs are widely used for: the development of different antibacterial package materials [1], wound healing applications [2,3], photocatalytic applications [4], wastewater treatment [5], different sensors [6], etc.

The most important properties of CS are the presence of reactive amino and hydroxyl side groups which through hydrogen or covalent bond can interact with TiO_2_ NPs and form nanocomposites with high concentration without NPs agglomeration [6,7].

It is worth noting that most nanocomposite applications are based upon the high effective surface area of the NPs. However, it is difficult to obtain a uniform dispersion of nanoparticles owing to their strong tendency of agglomeration for their high surface energy. Agglomeration of NPs worsens the properties of nanocomposites due to the restriction of surface area [8,9]. In composites based on TiO_2_ NPs, agglomeration affected their photocatalytic activity [3,10], the conductivity of ion batteries [11], the biological activity of nanocomposites with TiO_2_ NPs [12], the antimicrobial capacity [1], the adsorption capacities [5], etc.

In recent years, it has become clear that the interface layer formed around nanoparticles in polymer nanocomposites is critical for controlling their dispersion (see, for example, [13,14,15,16,17]). This layer appears when strong molecular interactions between nanoparticles and polymer matrix are present; it occupies a significant volume fraction of the polymer matrix and it exhibits properties that differ from both polymer matrix and NPs [8,10,18]. As a result of these strong molecular interactions, most properties of the nanocomposite significantly depend upon the interphase layer (see, for example [8,9,18,19,20,21,22,23,24,25,26,27]). The experimental and theoretical investigations in the recent years provided a large amount of information about interphase and interfacial interactions in polymer nanocomposites; nevertheless, they focused on dynamic of polymer chains [22,23,24,25], glass transition temperature [26], mechanical properties [8,20,21], adsorption of polymer chains on the nanoparticle surface [27]. However, to the best of our knowledge, there are no publications in the literature that optimizes the concentration of NPs before they agglomerate (or nanocomposites with the largest NP surface area) by considering the effect of the interphase layer. Additionally, there are no reported investigations between of the interface layer in CS–TiO_2_ nanocomposites.

It is worth noting that chitosan is dielectric material and TiO_2_ is a high resistivity semiconductor (ca. 10^15^ Ohm cm) with dielectric constant ca. 100 which dependents upon the polymorphism of NPs [28]. Therefore, dielectric spectroscopy can be advantageously used to determine relevant parameters of the interface layer and the highest concentration of NPs before they agglomerate (decreasing the effective surface area).

Recently, some publications on the use of TiO_2_ NPs for the development of an electrochemical sensor for the detection of pesticides, including imidacloprid (IMD) are reported [29,30]. IMD is one of the most used neonicotinoids for the crop protection worldwide due to its low soil persistence and high insecticidal activity at a very low application rate [31]. IMD is absorbed by plants via either their roots or leaves and then is transported throughout the tissues of the plants [32]; it may be present in various foods that ultimately affect the human health [29].

Different methods or techniques have been used for the detection and quantification of this insecticide in wastewater. High performance liquid chromatography, gas chromatography–mass spectrometry, liquid chromatography mass spectrometry and optical technique are commonly used methods for the detection of IMD. These instrumental methods are accurate but expensive and time consuming, requiring lengthy sample extraction and cleanup procedures [29,33,34]. On the other hand, nitro-group of IMD can be reduced electrochemically at a negative potential. Therefore, due to electro-activity of IMD, several electrochemical methods were reported for the detection of IMD using different electrodes modified by polymer–NPs composites [30]. The electrochemical sensor and biosensor platforms have emerged as powerful analytical methods to detect pesticides due to ease of detection and appreciable sensitivity [35]. However, to the best of our knowledge, there are no reports about the electroanalytical determination of IMD using electrodes modified by CS–TiO_2_ NPs nanocomposites. Due to interaction of CS with TiO_2_ NPs and the formation of interface layer which prevents the agglomeration of NPs at relative high concentration, we propose that such modified electrode will offer high surface area and enhance the accessibility of IMD to the recognition NPs sites. Thus, we expect that the agglomeration will be affected on sensibility of CS–TiO_2_ sensor for detection of imidacloprid.

Based upon the above, this work aims to investigate the properties of CS–TiO_2_ films as a function of the concentration of NPs by considering an interface layer formation, and to shed light about the development of a sensor for electroanalytical detection of IMD.

## 2. Materials and Methods

Chitosan (CS), medium molecular weight (ca. 350 kDa), deacetylation ca. 72%, acetic acid (99.7%), and TiO_2_ NPs with dimension between 20–40 nm was purchased from Chemours Co. (Mexico City, Mexico), and used as received.

### 2.1. Preparation of Films

The films were cast by dissolving 1% *w*/*w* of chitosan in 1% *w*/*w* aqueous acetic acid solution with continuous magnetic stirring for 24 h. Different concentration of TiO_2_ nanoparticles were dispersed within the chitosan solution for 3 h. Then, each solution with different concentration of TiO_2_ nanoparticles was placed on the ultrasonic tip for 10 min with intervals of 3 min and pulses of 5 s to avoid heating the solution. This solution was poured into a Petri dish and allowed to evaporate at 60 °C for 18 h to obtain the chitosan acetate films. The neutralization of acetate film was done with aqueous ammonia solution 2 mol/dm^3^. The initial pH was ca. 3.7–4. For impedance measurements, CS–TiO_2_ films were gold-sputtered on both sides to serve as contacts.

### 2.2. Preparation of Working Electrode

The glassy carbon electrodes (GCE) were hand polished with 0.3 and 0.1 µm alumina slurries, washed under ultrasound for 5 min with ethanol, and dried in air oven for 2 h at 60 °C. An aliquot of 7 µL of each chitosan–TiO_2_ solution prepared for films was taken. This aliquot was deposited on the surface of each GCE. The electrodes were dried in air for 3 h and a uniform film was formed over each electrode surface. In total, 6 electrodes were modified with CS–TiO_2_ at each TiO_2_ concentration, 1 electrode modified with CS, and the GCE electrode used as a target.

### 2.3. Characterization

The crystalline structure of TiO_2_ NPs and CS–TiO_2_ films were tested by an X-ray diffractometer (Rigaku Dmax 2100, Austin, TX, USA) with Cu Kα radiation (λ = 0.154 nm).

The interaction between CS functional groups with TiO_2_ NPs was analyzed by FTIR measurements on a Perkin Elmer Spectrum GX spectrophotometer using ATR (MIRacle™, Madison, WI, USA) sampling technique, with a diamond tip, in the range from 4000 to 1000 cm^−1^ at room temperature.

The amount of free water was determined by thermogravimetric analysis (TGA) (TGA 4000—PerkinElmer; PerkinElmer, Inc., Waltham, MA, USA). Measurements were made in the dry air with a heating rate of 10 °C/min.

Impedance measurements were carried out using Agilent 4249 A in the frequency range 40 Hz–100 MHz with an amplitude of AC voltage 100 mV at room temperature. The dielectric constant of nanocomposites has been calculated from impedance spectra using ZView program from the following relationship: ε = (C·d)/(ε_0_·S), where C is capacitance measured at the frequency 100 Hz, *d* and *S* are the thickness and area of samples, respectively. Film thickness was measured in each sample using micrometer Mitutoyo with resolution 1 mkm.

Voltammetry cyclic measurements were performed using a potentiostat–galvanostat VoltaLab PGZ301 (Radiometer analytical), and a conventional three-electrode cell at room temperature. The CS/TiO_2_ modified GCE electrodes were used for electrochemical experiments as working electrode, Ag/AgCl electrode was used as reference electrode and platinum plaque was used as a counter electrode. The cell electrochemical was covered with black tape to avoid photocatalytic effect. Na_2_SO_4_ 0.05 M was used as electrolyte and the imidacloprid concentrations were of 10, 50, 100, 250 and 500 ppm, this last concentration according with the imidacloprid solubility. The potential range was established between −1.8 and −0.1 V, scan rate of 150 mV/s and, prior to use, the working electrode was stabilized through 10 continuous repetitive cyclic voltammograms running, to obtain a stable and reproducible background current. All experiments were carried out with solutions previously deaerated with a stream of N_2_ gas bubbled in the solution for 10 min.

## 3. Results and Discussion

### 3.1. Morphology

According to SEM measurements the most of NPs have dimension between 18 and 38 nm. SEM images recorded from of CS–TiO_2_ nanocomposites films with 10 wt.% and 30 wt.% show homogeneous distribution of NPs (Figure 1a,b) in CS matrix. In contrast, the films with 50 wt.% agglomeration is observed (Figure 1c). This behavior plays an important role in the explaining structural properties of nanocomposites and their application.

### 3.2. XRD

Figure 2 shows XRD patterns of TiO_2_ NPs and CS–TiO_2_ NPs composite film with 40 wt.% of NPs. The XRD pattern of pure TiO_2_ NPs showed diffraction peaks and confirm the presence of anatase (JCPDS PDF#21-1272) and rutile phases (JCPDS PDF#21-1276) which often observed in TiO_2_ powder (see, for example [36,37]). In the pattern of CS–TiO_2_ films, additionally to diffraction peaks of anatase and rutile observed broad peak related to amorphous CS (between 8–18 degree).

The quantification and refinement of the volume fraction of every crystalline phase was carried out by the Rietveld method implemented in the Fullprof software; this allows to calculate the volume fraction of every phase (68.5% of anatase and 31.5% of rutile). This estimation is important for the appropriate interpretation of dielectric measurements (see Section 3.5).
Figure 2XRD pattern of TiO_2_ NPs and CS–TiO_2_ film with 40 wt.% of NPs.
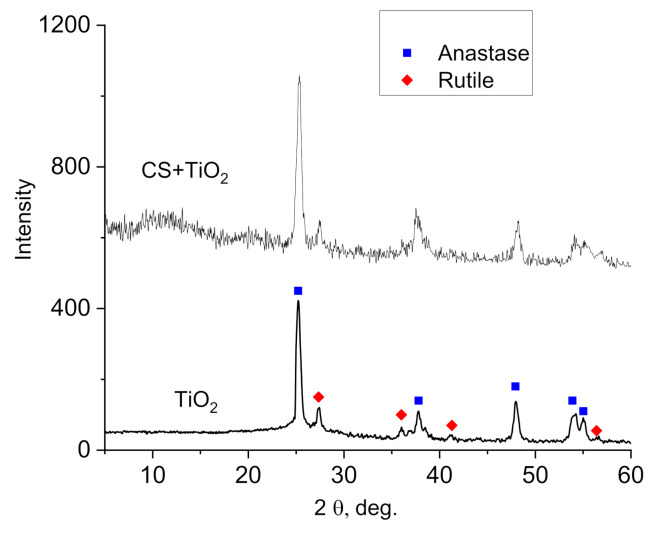


### 3.3. FTIR

FTIR spectroscopy was used to observe the interactions between the chitosan and TiO_2_ NPs. In the case of neat CS, the spectrum shows the characteristics bands reported by other authors [38,39]: the broadband characteristic peak centered at 3322 cm^−1^ correspondents to the overlap of stretching vibration of -NH and -OH groups of CS, and peak at 1562 cm^−1^ related to bending vibration of NH_2_ group. In CS–TiO_2_ films, these peaks shift from 3322 cm^−1^ and 1562 cm^−1^ to 3269 and 1547 cm^−^^1^ wavenumbers (Figure 3).

The shift of characteristic peaks in FTIR measurements are due to the interaction between CS reactive amino and hydroxyl groups with TiO_2_ NPs. Other studies have proposed a different mechanism of interaction including chelation [7,40,41]; the formation multiple coordination bonds between organic molecules and metals; interaction of CS side groups with Ti ions on TiO_2_ surface [42]; covalent interaction between CS and TiO_2_ [43]; electrostatic interaction between negative charge of CS carboxyl groups and TiO_2_ positive charge [44].

Another possible mechanism of the interaction between CS and NPs can be related to the dissociation of water on the surface sites or defects of TiO_2_ and the formation of two different OH functional groups at the surface of NPs [45,46]. These groups are responsible to interaction of TiO_2_ with lateral groups of CS. The most important thing for us is the presence of interaction between CS and TiO_2_ NPs, which is responsible for the formation of an interface layer surrounding NPs. The properties of this interphase layer will be discussed in the next section. In addition, a decrease in the intensity of the band centered at 3322–3269 cm^−1^ demonstrates a decrease in the water absorption capacity with an increase in the concentration of NPs in the films (decrease in intensity).
Figure 3FTIR spectra of neat CS film and CS–TiO_2_ films with 40 and 60 wt.% of NPs.
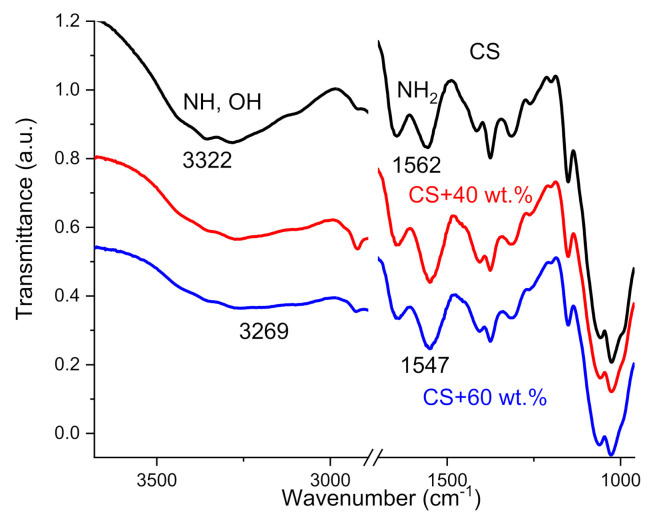


### 3.4. TGA

The interaction between CS side groups with TiO_2_ NPs precludes water absorption capacity and it is supported by TGA measurements (Figure 3). Water absorption in CS is closely linked to the availability of amino and hydroxyl groups of CS interact via hydrogen bond with water molecules [47,48]. It was shown that the water content depends upon TiO_2_ NPs concentration (Figure 4) and decreases with increasing weight % of NPs (10.9% in neat CS and 7.1% in CS–TiO_2_ film with 60 wt.% of NPs, at the temperature 150 °C). As it was shown by FTIR analysis, CS lateral groups can be bond with OH groups on the surface of NPs that is responsible for decreasing of water content in the nanocomposite films.

### 3.5. Dielectric Measurements

Due to the difference between dielectric constant of CS matrix and TiO_2_ NPs, the dielectric spectroscopy measurements can be used for the determination of parameters of the interface layer and the highest concentration of NPs before they agglomerate (decreasing effective surface area). 

In the literature have been proposed a three-phase model to describe the dielectric properties of polymer–dielectric NPs composites [49,50]. Here, the effective dielectric constant of such composite materials depends upon the ε_m_ of the polymer matrix, the ε_NPs_ of NPs, and the ε_int_ of interface layer between filler and the dielectric matrix. To describe such three-phase system, Refs. [49,50] introduce a parameter *K*, termed the interface volume constant, which accounted for the matrix–filler interaction strength as: (1)$$\emptyset_{int}=K\emptyset_{NPs}\emptyset_{m}$$
where *Φint*, *Φ_NPs_* and *Φ_m_* are the volume fractions of interface phase, dielectric particles and polymer matrix, respectively. K depends upon the degree of particle clustering. A value of zero for K indicates that there is no interracial phase region between the NPs and polymer matrix. Positive values for K are related to an interaction between the polymer and NPs.

This model has demonstrated that the dependence of the dielectric constant on NPs concentration is nonmonotonic and in dependence of interface volume fraction as function of NPs concentration can observed extremum due to an overlap of interface layers and NPs agglomeration.

Before fitting, experimental dependence of effective dielectric constant of nanocomposite is necessary to recalculate the weight fraction of TiO_2_ NPs (*W_t_*) in their volume fraction *Φ_NPs_* using next equation [51]:(2)$$\emptyset_{NPs}=\frac{W_{t}}{W_{t}+\left(\frac{\rho_{TiO2}}{\rho_{CS}}\right)\left(1-W_{t}\right)}$$
where, *ρ_TiO_*_2_ and *ρ_CS_* denote the *TiO*_2_ and *CS* density.

The density of CS films is ca. 1.5 g cm^−3^ [52,53]. TiO_2_ NPs, according to XRD measurements, have two phases (68.5% of anatase with density *ρ_anat_* = 3.78 g/cm^3^ and 31.5% of rutile with density *ρ_rut_* = 4.23 g/cm^3^) [54]. Using simple mixture rule, the density of TiO_2_ NPs can be calculated [51]: (3)$$\rho_{TiO2}=0.685*\rho_{anat}+0.315\rho_{rut}=3.92\;g/{\mathrm{cm}}^{3}$$

Next important value for fitting experimental dependence is effective dielectric constant of TiO_2_ NPs (*ε_NPs_*). This effective dielectric constant of TiO_2_ NPs can be estimated using Lichtenecher logarithmic model [55,56]:(4)$$\log\epsilon_{NPs}=\emptyset_{anat}\log\varepsilon_{anat}+\emptyset_{rut}\log\varepsilon_{rut}$$
where *Φ_anat_* and *Φ_rut_* represent the volume fraction of anatase and rutile, respectively. 

Taking data *ε_anat_* = 75 and *ε_rut_* = 105 [56], the effective dielectric constant of TiO_2_ NPs is equal 82.5.

This value well correlates with dielectric constant (ca. 100) measures at the frequency 100 Hz on pellets consist of TiO_2_ NPs [57,58,59]. Therefore, dielectric constant of nanocomposites has been measured at the frequency 100 Hz. 

In this work, we fit the model of effective dielectric constant
(5)$$\varepsilon =f\left(\varepsilon_{NPs},\varepsilon_{int},\varepsilon_{m},K,\phi_{NPs}\right)$$
proposed in Refs. [35,36] to experimentally obtained values of effective dielectric constant of CS–TiO_2_ composite ${\left(\epsilon_{i}\right)}_{i=1}^{N}$. The least squares fitting is performed using a standard function of genetic algorithm optimization in the Scilab [60]. The fitting error is specified as the sum of the squares of the differences between the dielectric constants predicted by the model and obtained in result of N measurements.
(6)$$E\left(K,\varepsilon_{int}\;\right)=\overset{N}{\underset{i=1}{\sum }}{\left|f\left(\varepsilon_{NPs},\varepsilon_{int},\varepsilon_{m},K,\phi_{NPs,i}\right)\;-\;\epsilon_{i}\right|}^{2},\;\;\;i=1\ldots N$$

The dependence of the fitting error *E*(*K*, *ε_int_*) on interphase volume constant *K* and interphase dielectric constant *ε_int_* is shown in Figure 5.

The graph shows that the function is not unimodal. This fact helped determine the selection of the genetic algorithm as an optimization method.

Dielectric constant of CS has been obtained from measurements on neat CS films. Dielectric constant of TiO_2_ was taken 82.5 (calculated using Equation (4)). Only the values of *K* and *ε_int_* are the adjustable parameters. In the process of fitting the program found the optimum values of *K* and *ε_int_* at the minimum value of fitting error. As a result of optimization, the fitted values are *K* = 15, *ε_int_* = 19.7 and fitting error *E*(*K*, *ε_int_*) = 1.25.

Figure 6 shows the dependencies of the dielectric constant obtained in CS–TiO_2_ films at the frequency 100 Hz with different NPs concentration (points). Results of the referred fittings are shown on Figure 5 as continuous line. One can observe that this three-phase model fits well with the experimental results. Positive values of K mean that there is significant interface in CS–TiO_2_ interactions. The value of interface dielectric constant is less than ε of neat CS. The low value of ε_int_ can be related to the interaction of CS lateral groups with OH groups on the surface of NPs that is responsible for the decreasing of water content in the interface layer. 

However, the most important result of fitting is shown on the insert of Figure 6: dependence of interface volume fraction of nanocomposite on TiO_2_ NPs volume fraction. This dependence demonstrates maximum at the concentration of TiO_2_ at ca. 39 wt.%. At higher concentration of NPs, interface volume fraction decreases due to agglomeration of NPs, which is well observed in SEM measurements (Figure 1c). Thus, it can be assumed that an electrode with CS–TiO_2_ films with 40 wt.% NPs will demonstrate better sensitivity to imidacloprid, since at a higher concentration, due to NPs agglomeration, the surface area of NPs will decrease.

### 3.6. Cyclic Voltammetry Measurements

Figure 7 shows the redox behavior voltamperograms of imidacloprid measured on the electrodes modified by CS–TiO_2_ nanocomposites films with different concentration of TiO_2_ NPs. 

For all redox behavior, the plot of peak current (Ip) versus the respective concentration of IMD was found to be linear in the range 10–500 ppm (3.9 × 10^−5^–2 × 10^−3^ mol/L) as shown in the insets in Figure 5. The cathodic peak can be observed at a value of −1.42 ± 0.048 V for GCE and −1.36 ± 0.03 V for the GCE/CS electrode. It can be observed that in the electrode of low concentration of TiO_2_ nanoparticles at 10 wt.%, it is evident that, during the cathodic scan, a single reduction peak was observed at a potential of −1.3 ± 0.025 V. The noted cathodic process is derived from the nitro-group irreversible reduction of imidacloprid [33,61,62,63]. In the case of the electrode GCE/CS–TiO_2_ 20 wt.% only the reduction of the IMD to 500 ppm was observed in −1.42 V and its oxidation value was presented for the lowest concentrations in −1.52 V. For electrodes with CS–TiO_2_ with 30, 40, 50 and 60 wt.% the reduction and oxidation of the imidacloprid was observed, this indicates the reversibility of the electrode in the behavior redox measurement. However, for electrodes with CS–TiO_2_ 30 wt.%, CS–TiO_2_ 40% wt.% and CS–TiO_2_ 50 wt.%, one can see the appearance of two peaks of reduction: one at −1.663 V and another at −1.351 V. This the second peak is clearly observed at concentrations of 50, 100 and 250 ppm of IMD. At the electrode with 40 wt.% of TiO_2_ nanoparticles, the second peak of reduction is more visible except for the low concentration of 10 ppm. This second reduction peak had already been reported by Navalon et al. [64] through of differential pulse polarographic method. Thus, it was able to propose an overall reduction mechanism for each of the two imidacloprid reduction peaks. For the first peak, the nitro group of the imidacloprid molecule takes four electrons to give the corresponding hydroxylamine derivative and then in the second reduction peak this compound takes two electrons to be transformed in the corresponding amine derivative. 

The limit of detection (LOD) and the limit of quantitation (LOQ) (Table 1) were calculated from using the equations: LOD = 3 sd/m and LOQ = 10 sd/m, where sd is the standard deviation of the intercept and m is the slope of the calibration curve [65].

The reported detection limit was in the same order for all electrodes and compared to other modified electrodes reported in the literature [33,65,66,67,68]. However, the electrode modified by CS–TiO_2_ film with 40 wt.% presents the best sensitivity to imidacloprid compare with another investigate (Table 2) modified electrodes because it detects two peaks of reduction, and its current values are higher above −150 µA. 

Hg(Ag) FE, Silver-amalgam film electrode; BDD, Boron-Doped Diamond; CPE, carbon-paste electrode; GCE, glassy carbon electrode; nAgnf/nTiO2nf/GCE, nanosilver Nafion^®^/nanoTiO2 Nafion^®^ modified glassy carbon electrode; PCz/CRGO/GCE, poly(carbazole)/chemically reduced graphene oxide modified glassy carbon electrode; BiFE, Bismuth-film electrode.

## 4. Conclusions

In this paper, for the first time, we propose the potential application of chitosan–TiO_2_ nanocomposites for the development of an electroanalytical sensor to detect imidacloprid with high sensitivity. Preliminary studies of imidacloprid detection show that the best results are obtained on a modified electrode with CS–TiO_2_ NPs with 40 wt.% of NPs. According to dielectric spectroscopy measurements on the nanocomposite, the dependence of the dielectric constant on NPs concentration is nonmonotonic. By using a three-phase model which includes: (1) CS matrix, (2) TiO_2_ NPs and (3) interface layer between NPs and CS matrix and by fitting dielectric spectroscopy measurements, the interface volume fraction was calculated. The highest calculated interface volume fraction is 39 wt.% of NPs which is practically the same value at which the best performance of the nanocomposite is observed. Our results strongly suggest that the interface layer is responsible for the good dispersion of NPs in chitosan matrix without agglomeration. At higher NPs concentration, the interface volume fraction decreases due to their overlap that leads to NPs agglomeration which ultimately decreases their effective surface area. TiO_2_ nanoparticles serve as molecular recognition sites for electroanalytical imidacloprid detection; therefore, their agglomeration decreases the sensitivity of imidacloprid detection.

The construction of such a sensor requires additional investigation. However, the methodology presented in this work, which allows to determine the optimum NPs concentration, may prove useful in the design and optimization of polymer-based nanocomposites for development nanocomposite for different applications. 

## Figures and Tables

**Figure 1 polymers-14-01686-f001:**
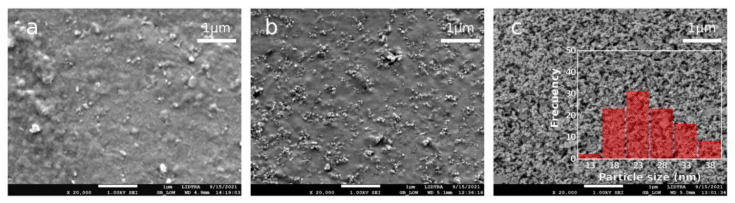
SEM images of CS–TiO_2_ nanocomposites films with (**a**) 10 wt.% of TiO_2_ NPs, (**b**) 30 wt.% of TiO_2_ NPs and (**c**) 50 wt.% of TiO_2_ NPs. Insert on Figure 1c shows NPs distribution.

**Figure 4 polymers-14-01686-f004:**
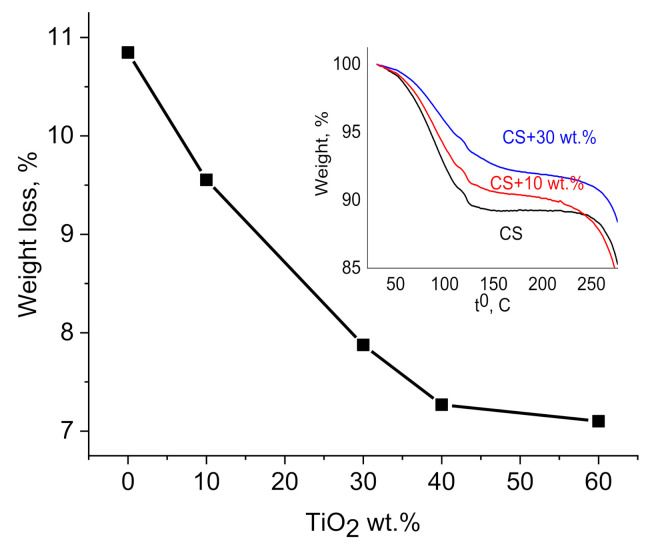
Dependence of weight loss in CS–TiO_2_ films on NPs wt.% measured at the temperature 150 °C. Insert shows TGA measurements of pure CS and CS–TiO_2_ NPs films with 10 and 30 wt.% of NPs.

**Figure 5 polymers-14-01686-f005:**
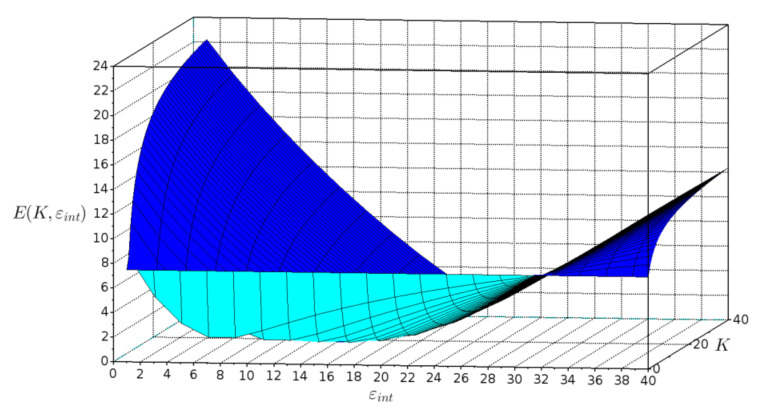
Plot of fitting error *E*(*K*, *ε_int_*) versus interphase volume constant *K* and interphase dielectric constant *ε_int_*.

**Figure 6 polymers-14-01686-f006:**
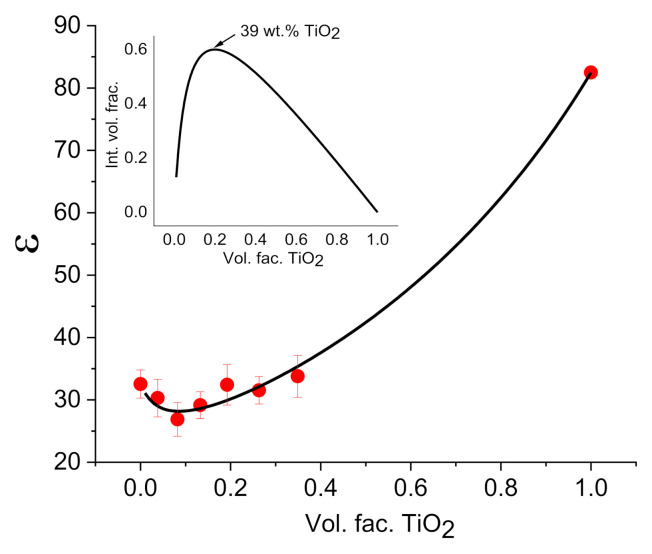
Dependence of the dielectric constant obtained in CS–TiO_2_ films at the frequency 100 Hz with different NPs concentration (points). Continuous line shows the results of fitting. Bars represent the standard deviation calculated from measurements on 4 samples. Insert shows the dependence of interface volume fraction on TiO_2_ volume fraction calculated using models proposed in [49,50].

**Figure 7 polymers-14-01686-f007:**
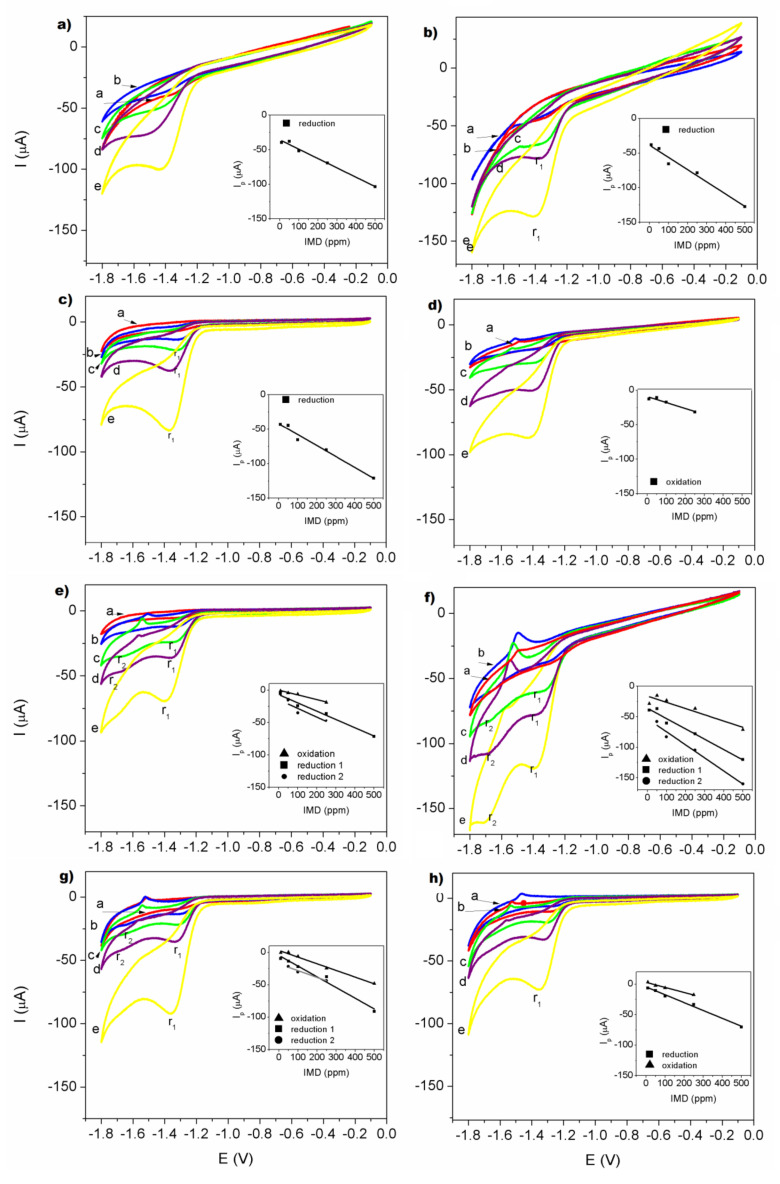
Cyclic voltammograms obtained for IMD on electrodes: (**a**) GCE; (**b**) GCE modified by CS; (**c**) GCE modified by CS-TiO_2_ film with 10 wt.% of NPs; (**d**) GCE modified by CS-TiO_2_ film with 20 wt.% of NPs; (**e**) GCE modified by CS–TiO_2_ film with 30 wt.% of NPs; (**f**) GCE modified by CS-TiO_2_ film with 40 wt.% of NPs; (**g**) GCE modified by CS–TiO_2_ film with 50 wt.% of NPs; (**h**) GCE modified by CS-TiO_2_ film with 60 wt.% of NPs. Measurements were carried out at 150 mVs^−1^ in 0.05 M Na_2_SO_4_ as supporting electrolyte, pH = 7. Each cycle corresponding to the different concentrations: a-10, b-50, c-100, d-250, e-500 ppm. Inserts show relationship between peak current of reduction and/or oxidation vs IMD concentration.

**Table 1 polymers-14-01686-t001:** Analytical parameters for the determination of IMD on different electrodes.

Electrode	Cathodic Peak (V)	Anodic Peak (V)	Linear Range (mol/L)	Cathodic			Anodic		
				LOD (mol/L)	LOQ (mol/L)	R^2^	LOD (mol/L)	LOQ (mol/L)	R^2^
GCE	−1.42 ± 0.048	----	3.9 × 10^−5^–2 × 10^−3^	1.82 × 10^−4^	6.08 × 10^−4^	98.5	---	---	---
GCE/CS	−1.36 ± 0.030	---	3.9 × 10^−5^–2 × 10^−3^	2.76 × 10^−4^	9.20 × 10^−4^	96.7	---	---	---
GSE/CS-TiO_2_ 10%	−1.37 ± 0.026	---	3.9 × 10^−5^–2 × 10^−3^	2.60 × 10^−4^	8.68 × 10^−4^	97	---	---	---
GSE/CS-TiO_2_ 20%	−1.42 ± 0.030	−1.52 ± 0.032	3.9 × 10^−5^–2 × 10^−3^	---	---	---	2.96 × 10^−4^	9.85 × 10^−4^	90.5
GSE/CS-TiO_2_ 30%	−1.34 ± 0.036	−1.51 ± 0.046	3.9 × 10^−5^–2 × 10^−3^	2.04 × 10^−4^	6.79 × 10^−4^	98.2	1.42 × 10^−4^	4.75 × 10^−4^	97.7
GSE/CS-TiO_2_ 40%	−1.34 ± 0.043	−1.52 ± 0.027	3.9 × 10^−5^–2 × 10^−3^	2.74 × 10^−4^	9.14 × 10^−4^	96.7	6.2 × 10^−4^	2.07 × 10^−3^	84.7
GSE/CS-TiO_2_ 50%	−1.32 ± 0.028	−1.54 ± 0.022	3.9 × 10^−5^–2 × 10^−3^	2.53 × 10^−4^	8.44 × 10^−4^	97.2	1.67 × 10^−4^	5.58 × 10^−4^	98.8
GSE/CS-TiO_2_ 60%	−1.30 ± 0.030	−1.52 ± 0.046	3.9 × 10^−5^–2 × 10^−3^	1.46 × 10^−4^	4.88 × 10^−4^	99.1	1.44 × 10^−4^	4.79 × 10^−4^	97.6

**Table 2 polymers-14-01686-t002:** Comparison of different electrodes in the detection of IMD.

Method	Electrode	Linear Range (mmol L^–1^)	LOD (mmol L^–1^)	LOQ (mmol L^–1^)	References
SWV	Hg(Ag) FE	3.55–185.6	1.05	3.6	[69]
SWV	BDD	30–200	8.6	28.6	[33]
DPV	CPE	6.7–117.4	2.04	6.8	[70]
DPV	*n*Ag*n_f_*/*n*TiO_2_*n_f_*/GCE	0.5–3.5	0.25	0.8	[66]
DPV	PCz/CRGO/GCE	3–10	0.44	1.5	[71]
DPV	BiFE	9.5–200	2.9	–	[72]
CV	GCE	10.9–1956	30.1	101.6	[62]
CV	CPE	1–7	0.63	2.1	[70]
CV	PCz/CRGO/GCE	3–10	0.22	0.7	[72]
CV	GCE/CS-TiO_2_ 40%	0.039–2	0.6	2.1	This work

## Data Availability

Not applicable.

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
