# Peer review of "Potential Use of Chitosan-TiO2 Nanocomposites for the Electroanalytical Detection of Imidacloprid"

_polymers, 2022, doi:10.3390/polym14091686_

Round 1

Reviewer 1 Report

Reviewer’s comments on the manuscript: Potential use of Chitosan-TiO2 nanocomposites for the electroanalytical detection of imidacloprid written by Blanca E. Castillo Reyes, Evgen Prokhorov, Gabriel Luna-Barcenas and Yuriy Kovalenko

The reviewed manuscript investigates the properties of CS-TiO2 films on the concentration of NPs by considering interface layer formation, and the possibility of the development of a sensor for electroanalytical detection of IMD. The manuscript presents very interesting data having high application potential. In my opinion the manuscript is in the journal’s fields of interests. Experiments are properly planned and the obtained data are clear. However, in my opinion some additional information should be added or clarified before publications process. Thus my suggestion is major revision.

Reviewers comments and suggestions:

  • Title: there is an editorial mistake in the title. It should be TiO2.
  • Abstract perfectly stays the relevance of the presented studies.
  • line 36: Please add some references.
  • Materials and methods: In my opinion the main weakness of this manuscript is very poor characterization of used chemical compounds. At least molecular weight and polydispersity index should be estimated. As far as TiO2 is concerned the crucial missing parameters are: specific surface area (BET), surface charge and particle size distribution. These parameters are obligatory for discussion.
  • line 87: There is no such compound as NH4OH, if you mean ammonia solution it should be: NH3 x H2O.
  • Please change “M” into “mol/dm3”.
  • Fig. 1: What was the magnification used?
  • Fig. 1: Please correct the figure caption. It should be specified what is a), b) and c) but you mentioned only a) and d) (???) in the caption.
  • line 154: there is an extra space. The same lines: 172, 179, 181.
  • Fig. 2: Why did you chose the concentration of 40% of NPs?
  • lines 167-168: please explain it more clearly. Chitosan can interact with chitosan?
  • The Authors should choose one style of presenting numbers with their units. I recommend number, space, unit, for example: 150º C. Please compare lines 185 and 190.
  • lines 197-255: There are not results, this part of text is rather introduction and it should be transferred.
  • There is a large inconsistency in the method of numbering the equations, sometimes only the number in parentheses is given, sometimes with a comma, and sometimes with a dot. Please unify this.
  • Fig. 5: caption is not meaningful. Please correct it.
  • Fig. 6: The error bars are missing.
  • lines 293-294: smaller front?
  • lines 335-337: I really appreciate this part.
  • Table 2: This table should be change into more readable form.
  • lines 356: the dot is missing.

Reviewer 2 Report

Journal: Polymers                    

The manuscript entitled “Potential use of Chitosan-TiO2 nanocomposites for the electroanalytical detection of imidacloprid” by Reyes, et al bought the electroanalytical aspects for the detection of hazardous insecticides. The current work aims to study the properties of chitosan-TiO2 9 nanocomposites in which nanoparticles with high surface area serve as molecular recognition sites for electroanalytical imidacloprid detection. The manuscript is interesting. The manuscript can be accepted after addressing the following comments.

Comments:

  1. Rewrite the preparation of the working electrode, seems this part is confusing.
  2. In many places sentences need revision.
  3. Scale bar is not clear in the SEM figures.
  4. Label the respective peaks in the FTIR figure
  5. On line 180, starting; the Authors stated that “This conclusion is supported by TGA measurements”. Which conclusion? Mention clearly.
  6. Voltammetry cyclic measurements should be written as “Cyclic voltammetry measurements”.
  7. Improve the quality of Figure 7.
  8. Some potential references have been reported for the on-site detection of pesticides. Those must be properly cited. Discuss these papers in the introduction section.

Polymers 2021, 13(23), 4110; https://doi.org/10.3390/polym13234110

Polymers 2021, 13(22), 3869; https://doi.org/10.3390/polym13223869

https://doi.org/10.1016/j.ccr.2021.214305

https://doi.org/10.1016/j.ccr.2021.214061

Round 2

Reviewer 1 Report

Dear Authors,

I do apreciate the changes made in the manuscript. Well done!

Reviewer

Reviewer 2 Report

The revised manuscript can be accepted for publication in polymers